

# House sparrows do not show a diel rhythm in double-strand DNA damage in erythrocytes

Emma Rosen, Lily Mikolajczak, Ursula K. Beattie and L. Michael Romero

Department of Biology, Tufts University, Medford, MA, United States of America

## ABSTRACT

DNA damage can be caused by a number of intrinsic and extrinsic factors. A recent study showed that free-living house sparrows *(Passer domesticus)* have higher DNA damage in the summer than the winter across five different tissues. This result was consistent when house sparrows were brought into captivity and exposed to comparable light cycles, with all other variables held constant. These results generated two hypotheses: (1) seasonal variation in DNA damage is related to circadian regulation and (2) seasonal variation in DNA damage is related to the total number of active hours. To investigate these hypotheses, we first quantified erythrocyte DNA damage in wild-caught house sparrows held in captivity on a 12L:12D light cycle at six points during the day to assess a diel or circadian rhythm but did not find one. We then performed a resonance experiment, in which birds experienced unnatural light cycles, and compared DNA damage in birds held on 6L:6D and 4.5L:7.5D resonance light cycles with their natural counterparts, 12L:12D and 9L:15D, respectively. We assessed corticosterone levels and DNA damage in blood before and after the resonance light cycles and DNA damage in abdominal fat, hippocampus, hypothalamus, and liver after the resonance light cycles. While our second experiment was not able to effectively test our hypotheses, we were able to demonstrate some interesting patterns. Throughout the resonance experiment, baseline corticosterone and testes size increased, consistent with the birds being photostimulated and preparing to breed. Surprisingly, the direction of change of DNA damage throughout the resonance photoperiod differed with tissue, which is not consistent with patterns during the breeding season in the wild. Our data indicate a potential uncoupling of the breeding physiology with the effect on DNA damage due to exposure to a resonance light cycle, which the birds may have interpreted as a skeleton photoperiod. Finally, though we were unable to fully disentangle the dynamics underlying seasonal DNA damage, we show that the previously documented patterns are not simply due to diel changes or the total amount of light exposure within a 24-hour period.

# INTRODUCTION

Measuring DNA damage may serve as a useful, integrative metric to assess an animal's physiological state, as it can be altered by both internal and external factors. If left unchecked, an accumulation of DNA damage can lead to changes in gene expression,

Corresponding author
Ursula K. Beattie,
ursula.beattie@tufts.edu

apoptosis, cancer, or shortened lifespan (reviewed in *Hakem, 2008*). *In vitro* studies have shown that double-strand breaks in DNA occur after just 10 min of either catecholamine or glucocorticoid exposure (*Flint et al., 2007*), a finding that has also been supported by *in vivo* studies in birds (*Beattie et al., 2024*; *Gormally et al., 2020*). On the whole-animal level, double-strand breaks can be induced by chronic stress (*Gormally et al., 2019b*), but this result is both context- and tissue-dependent (*Beattie et al., 2024*). These effects are not limited to double-strand breaks—glucocorticoid exposure and chronic stress can also shorten telomeres (*Casagrande et al., 2020*; *Herborn et al., 2014*).

A recent study has found a large effect of season on double-strand breaks—across five tissues, there was higher DNA damage in house sparrows (*Passer domesticus*) in the summer than in the winter (*Beattie et al., 2022a*). Even more surprisingly, this pattern persisted when wild house sparrows acclimated to captivity were exposed to summer or winter light cycles, with all other variables held constant (*e.g* temperature, food availability, humidity, lighting source, *etc.*). It is unlikely that this result was simply due to reproduction as house sparrows rarely breed in captivity. Similar studies in humans have suggested that this increase in DNA damage in the summer could be due to an increase in solar radiation (*Dušinská et al., 2002*; *Gerić et al., 2018*; *Møller et al., 1998*; *Verschaeve et al., 2007*), however this explanation is unlikely to explain differences in DNA damage in deep tissues under layers of feathers, and the birds in the captive study described above were not subjected to solar radiation. We suggest two potential hypotheses for this seasonal pattern of DNA damage. (1) Seasonal variation in DNA damage is caused by circadian regulation of the damage. In other words, seasonal fluctuations in damage are intrinsic to the circadian, and by extension circannual, cycle. (2) Rather than endogenous circadian mechanisms, DNA damage is a natural result of more time spent active. The increased active time that accompanies longer light cycles is then driving seasonal differences in DNA damage.

To our knowledge, diel rhythms of DNA damage have not yet been studied, but many factors than can alter levels of DNA damage do show rhythms. For example, both baseline corticosterone, which can increase DNA damage (*Flint et al., 2007*), and melatonin, which has antioxidant properties that could potentially protect against DNA damage (*Liu et al., 2013*; *Mir et al., 2022*), are both highest during the dark period, or scotophase (*Hau & Gwinner, 1994*; *Rich & Romero, 2001*). The dominant antioxidant in birds is uric acid (*Stinefelt et al., 2005*), which also shows a diel rhythm, but not in a consistent direction. In red legged partridges (*Alectoris rufa*), uric acid is higher during the scotophase (*Rodríguez, Tortosa & Villafuerte, 2005*), but in Japanese quail (*Coturnix japonica*), it is highest during the photophase, or the light period (*Herichová et al., 2004*). Because of these conflicting studies, we did not have an *a priori* prediction for when in the daily cycle DNA damage or uric acid would be highest, or if there would be a daily rhythm at all, in house sparrows. Furthermore, we chose to measure regulators of DNA damage in addition to the damage itself because DNA damage can be caused by a number of different factors—if patterns in DNA damage are correlated with upstream regulators, we can begin to tease apart mechanism.

In addition to directly testing for a daily rhythm of DNA damage, we also performed a resonance experiment, in which birds experienced unnatural light cycles. The total amount

of light in a 24-hour period would be controlled for, however it would be broken up into two light/dark cycles. For example, a 6L:6D photoperiod would be compared against a 12L:12D photoperiod because they have the same amount of light in a 24-hour period (12 h), but a different number of light/dark cycles (2 *versus* 1, respectively). If the birds in the resonance light cycles (6:6 or 4.5:7.5) have the same amount of DNA damage as the birds in the respective natural light cycles (12:12 and 9:15), then this would support the hypothesis that the seasonal pattern of DNA damage is due to the amount of active hours (hours of daytime, for this diurnal species). If the resonance birds had less DNA damage than their natural counterparts, we might conclude that circadian regulation is playing a part in the seasonal pattern of DNA damage.

## MATERIALS & METHODS

Animals were collected under a Massachusetts state collection permit. All experiments were conducted with approval from the Tufts Institutional Animal Care and Use Committee and comply with the Guidelines for Use of Wild Birds in Research (*Fair et al., 2023*). We tracked body weight weekly throughout the following experiments. If an animal dropped below 85% of its pre-experiment body weight or showed any other signs of declining health, it would be removed and allowed to recover, however no animals required removal.

### Experimental design–diel rhythms

Ten free-living house sparrows (2F:8M) were caught in September of 2023 in Eastern Massachusetts using mist nets. This sample size was based off of a power analysis that indicated a 95% chance of detecting a 50% difference in means. Birds were brought into captivity at Tufts University, housed in cages (45 cm × 37 cm × 33 cm), and allowed to acclimate to captivity for at least 2 weeks before experiments began. Because there were more males than females, six males were singly-housed and two males were doubly-housed with the two females (two sets of F/M pairs). Previous work has shown no effect of sex on DNA damage in house sparrows (*Gormally et al., 2019b*; *Gormally et al., 2020*). Furthermore, there are no differences in corticosterone between singly-housed and doubly-housed house sparrows (personal communication). During acclimation and for the duration of the experiment, food (commercially available mix of sunflower seeds and millet), grit, and water were provided *ad libitum*. All birds experienced a 12L:12D light cycle (lights on at 08:00, lights off at 20:00), which was approximately the outside light cycle at the time of capture. The average temperature in the bird rooms was 22 ± 0.3 °C and the average light intensity during the photophase was 232.9 ± 5.6 lux and 0 lux during the scotophase. All blood samples were collected by venipuncture of the alar vein using a 26-gauge needle within three minutes of entering the room (*Romero & Reed, 2005*). We took blood samples of approximately 10 µL at four-hour intervals over the 24-hour daily cycle (00:00, 04:00, 08:00, 12:00, 16:00, 20:00), however sampling was spaced out such that no samples were taken within 48 h of another. To take samples during the dark hours, researchers wore headlamps with blue filter paper (Roscolux Night Blue) covering the light, as blue light does not trigger the avian deep-brain photoreceptors that regulate circadian rhythms (*Wyse & Hazlerigg, 2009*). At each timepoint, we quantified double-strand DNA damage and uric

acid from whole blood. We measured uric acid from approximately 3 $\mu$L of whole blood using the UASure point-of-care device (Cat. No. U3003) (*Beattie et al., 2022b*), which has a range of up to 20 mg/dL. One out of 58 samples fell above the range of detection and was assigned a ceiling value of 20 mg/dL. 2 $\mu$L of whole blood was frozen in microcentrifuge tubes at $-80\ °C$ until assaying (*Gormally et al., 2020*). We employed a repeated-measures design in this experiment, such that each animal served as its own control. At the end of the experiment, remaining birds were used for other experiments in the lab (not part of the present study).

**Experimental design–resonance light cycles**
Eighteen wild male house sparrows *(Passer domesticus)* were caught in August (6L:6D) and October (4.5L:4.5D), brought into captivity, and allowed to acclimate as described above. The birds caught in August were used in the 6L:6D treatment group and the birds caught in October were used in the 4.5L:4.5D treatment group. All birds were initially on a 12L:12D light cycle during acclimation, but were then shifted to their treatment light cycles (6L:6D and 4.5L:7.5D, $n = 9$ each). We did not directly measure activity, but we observed that the birds were active in their respective photophases and inactive in their scotophases. This observation is also supported by previous work (*Binkley & Mosher, 1985*). The 6L:6D birds were compared to birds on a 12L:12D light cycle (data extracted from *Beattie et al., 2022a*), as they received the same amount of light in a 24-hour period (Fig. 1). Similarly, the 4.5L:7.5D birds were compared to the 9L:15D birds in the previous study (Fig. 1; *Beattie et al., 2022a*). In the wild, house sparrows in Eastern Massachusetts experience day lengths as long as 15 h in the summer and as short as 9 h in the winter. We purposefully chose not to sample birds experiencing 15 h of light to avoid confounding factors related to reproduction. Birds from the previous study *Beattie et al. (2022a)* had a longer acclimation period, but previous work shows that increases in DNA damage (*Gormally et al., 2019b*) and corticosterone (*Fischer, Wright-Lichter & Romero, 2018*) stabilize by two weeks of captivity, with no differences due to season of capture (*Lattin et al., 2012*). Birds were bled the day before resonance light cycles began for an initial blood DNA damage and corticosterone quantification. They then remained on the new light cycles for four weeks, at which point the birds were anesthetized with isoflurane and sacrificed for tissue collection.

We analyzed DNA damage in five tissues –blood, abdominal fat, hippocampus, hypothalamus, and liver, because all five have previously documented seasonal changes in DNA damage (*Beattie et al., 2022a*) and are major targets of glucocorticoid action. All tissue samples were stored on ice in 1.5 mL microcentrifuge tubes containing 1 mL of phosphate buffered saline (PBS; $Ca_{2+}$, $Mg_{2+}$ free). Prior to being placed in the tube with PBS, liver samples were placed in a weigh boat filled with chilled PBS for ~1 min to remove excess blood. Blood, abdominal fat, hippocampus, hypothalamus, and liver were immediately assayed for DNA damage and corticosterone. Testes were immediately weighed to obtain wet mass and then discarded.

**Comet assay**
Double strand DNA damage was quantified using a neutral comet assay (*Gormally et al., 2020*; *Olive & Banáth, 2006*). A total of 2 $\mu$L of whole blood was added to 800 $\mu$L of chilled

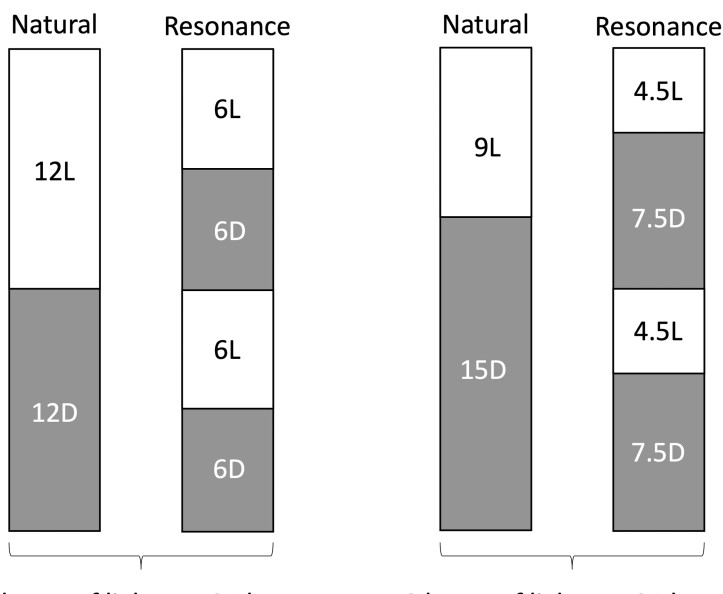

**Figure 1 Schematic representation of resonance experiment treatment groups.** Photophase, or light period of the day, is represented by white rectangles and scotophase, or dark period of the day, is represented by grey rectangles. Numbers inside rectangles represent the length of each period in hours.

PBS, followed by two five-fold dilutions (100 μL of suspension into 400 μL of fresh chilled PBS). Abdominal fat samples were prepared by mincing the tissue in the original 1 mL of PBS, followed by two four-fold dilutions with chilled PBS (250 μL of suspension into 750 μL PBS). Hippocampus samples were minced in the original 1 mL of PBS, followed by a 1:1 dilution (500 μL of suspension into 500 μL of PBS). Hypothalamus samples were minced in the original 1 mL of PBS and no further dilutions were conducted. Liver samples were minced in the original 1 mL of PBS, followed by a five-fold dilution (100 μL of suspension into 400 μL of PBS) and two 1:1 dilutions (250 μL of suspension into 250 μL of PBS).

We added 30 μL of each final sample suspension into 300 μL low-melting agarose (R&D Systems Cat. Minneapolis, MN, USA No. 425050002) warmed to 37 °C. After this dilution, samples were lightly vortexed and 30 μL were plated *via* pipette onto a CometSlide (R&D Systems Cat. No. 425202K01). Samples were plated such that the gel spread evenly throughout the well. Samples were sorted randomly and plated in duplicate. In addition, commercially available standardized control cells (R&D Systems Cat. No. 4257010NC) were plated on each plate to be run with unknown samples during each assay. Gels were allowed to solidify at 4 °C for 30 min. Following solidification, the slides were treated for 1 h at 4 °C with cold lysis buffer (R&D Systems Cat. No. 425005001) and dimethyl sulfoxide at a 10:1 ratio. The slides were then equilibrated in cold electrophoresis buffer (300 mM sodium acetate, 10 mM Tris base, pH 10) for 30 min at 4 °C and then electrophoresed in fresh electrophoresis buffer for 30 min (21V, 4 °C). Slides were washed twice in cold dH$_2$O and once in 70% ethanol for 5 min each. Finally, slides were dried in a 37 °C drying oven and then stored with a desiccant in a cool, dark place until staining and imaging.

The entirety of the comet assay was performed under low light conditions to prevent artificially increasing levels of DNA damage (*Beattie et al., 2022a*; *Gormally et al., 2020*; *Olive & Banáth, 2006*).

We then stained slides with SYBR™ Gold (Invitrogen, Waltham, MA, USA Cat. No. S11494) for 30 min, removed excess stain by washing with chilled $dH_2O$, and dried the slides at 37 °C. After staining, slides were imaged using a fluorescence microscope with a green fluorescent protein filter at x10 objective. The Fiji plugin, OpenComet was used to quantify damage in the microscope images (*Gyori et al., 2014*). Incorrect detections of comets were removed manually. The output metric used for further analysis was "TailMoment" for consistency with the natural light cycle data from *Beattie et al. (2022a)*.

## Corticosterone quantification

Whole blood and plasma were separated by centrifugation and plasma was stored in microcentrifuge tubes at −20 °C until assaying for corticosterone. We quantified corticosterone using an enzyme immunoassay (Arbor Assays, Ann Arbor, MI, USA Cat. No. K014H5W) according to the manufacturer's protocol (50 µL format). We used 5 µL of each plasma sample for the assay, which was diluted 1:50 with dissociation buffer (5 µL) and assay buffer (240 µL) (*Martin et al., 2017*). 50 µL of diluted plasma was plated into assay wells. There was no extraction step in this assay. Two assays were run and samples were run in duplicate, with treatments split across the assays. Intra- and inter-assay variability was 4.1% and 13.3%, respectively.

## Statistics

All statistics were conducted in R (version 4.2.1). Levene's test was used to assess homogeneity of variances (car package; *Fox & Weisberg, 2011*) and no data required transformation. In the diel rhythm experiment, both uric acid and DNA damage data were repeated measures, so we were able to use a linear mixed-effect model (lme4 package; *Bates et al., 2015*) with time of day as a fixed effect and bird identity as a random effect to control for individual differences. There was a significant effect of time for uric acid, so we performed a Tukey post-hoc analysis for pairwise comparisons (ghlt function, multcomp package; *Hothorn, Bretz & Westfall, 2008*).

In the resonance experiment, tissue sampling required terminal procedures, so the birds in each treatment group were unique and thus we could not use a repeated-measures statistical approach. DNA damage was analyzed using an *F* test with tissue, total light (9 or 12), and light cycle type (resonance or natural) as factors and total light as the slice term (*Bae, 2024*). We also included every interaction in the model. Because the model was significant in both birds with 9 and 12 total hours of light, we broke the model into two sub-models, one for 9 h of total light and one for 12 h of total light. The sub-models had tissue and light cycle type as factors and tissue as the slice term. We again included the interaction in the model. To compare testes weights between the two resonance groups, we used a student's *t*-test.

Sampling for weight, baseline corticosterone, and blood DNA damage is not terminal, so for those metrics, we were able to collect data before and after the birds experienced

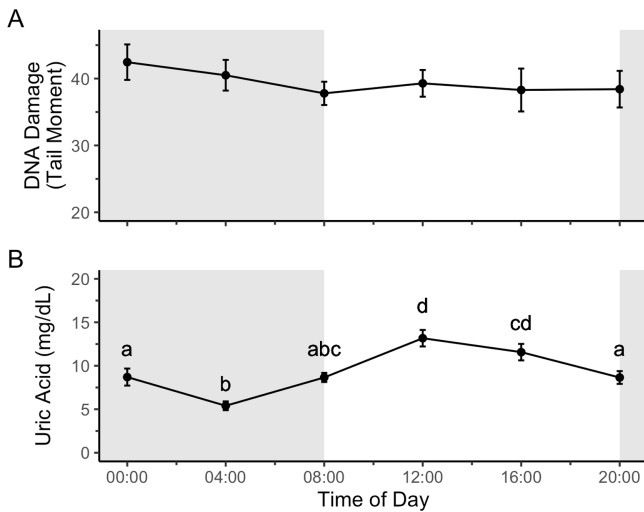

**Figure 2** **Uric acid, but not DNA damage, shows a diel rhythm in house sparrows on a 12:12 light cycle.** Different letters in panel B indicate significant post-hoc differences between timepoints. Shaded region represents the scotophase, while the white background region represents the photophase. $N = 10$ birds (2F:8M) and plotted values are means ± standard error.

resonance light cycles and use repeated-measured statistical techniques. We used linear-mixed effect models with time of sampling (pre- or post-resonance) and light cycle (6:6 or 4.5:7.5) as fixed effects and bird identity as a random effect.

## RESULTS

### Diel rhythms

Time of day did not significantly affect DNA damage (Fig. 2A; $F_{5,43} = 0.63$, $p = 0.67$), and thus double-stranded DNA damage does not show a diel rhythm. Uric acid, however, did show a diel rhythm (Fig. 2B; $F_{5,43} = 13.05$, $p = 9.21 \times 10^{-8}$) with lower blood uric acid levels during the scotophase and higher levels during the photophase.

### Resonance light cycles

The overall slice model indicated a significant interaction between light cycle type and tissue on DNA damage when accounting for total light (Fig. 3) in birds with 12 total hours of light ($F_1 = 76.66$, $p = 8.11 \times 10^{-15}$) and 9 total hours of light ($F_1 = 9.05$, $p = 0.003$). In birds with 12 total hours of light, this interaction is being driven by the fact that the 6L:6D birds had lower DNA damage than the 12L:12D birds in blood, abdominal fat, and hippocampus (blood: $F_1 = 15.04$, $p = 0.0003$; abdominal fat: $F_1 = 32.04$, $p = 4.50 \times 10^{-7}$; hippocampus: $F_1 = 14.27$, $p = 0.0004$), but not in hypothalamus or liver (hypothalamus: $F_1 = 2.71$, $p = 0.11$; liver: $F_1 = 2.51$, $p = 0.12$). Conversely, in birds with 9 total hours of light, this interaction is being driven by lower DNA damage in the 4.5L:7.5D than the 9L:15D birds in hypothalamus and liver (hypothalamus: $F_1 = 6.61$, $p = 0.01$; liver: $F_1 = 11.63$, $p = 0.001$), but not in blood, abdominal fat, or hippocampus (blood: $F_1 = 0.41$, $p = 0.52$; abdominal fat: $F_1 = 0.01$, $p = 0.92$; hippocampus: $F_1 = 1.97$, $p = 0.16$).

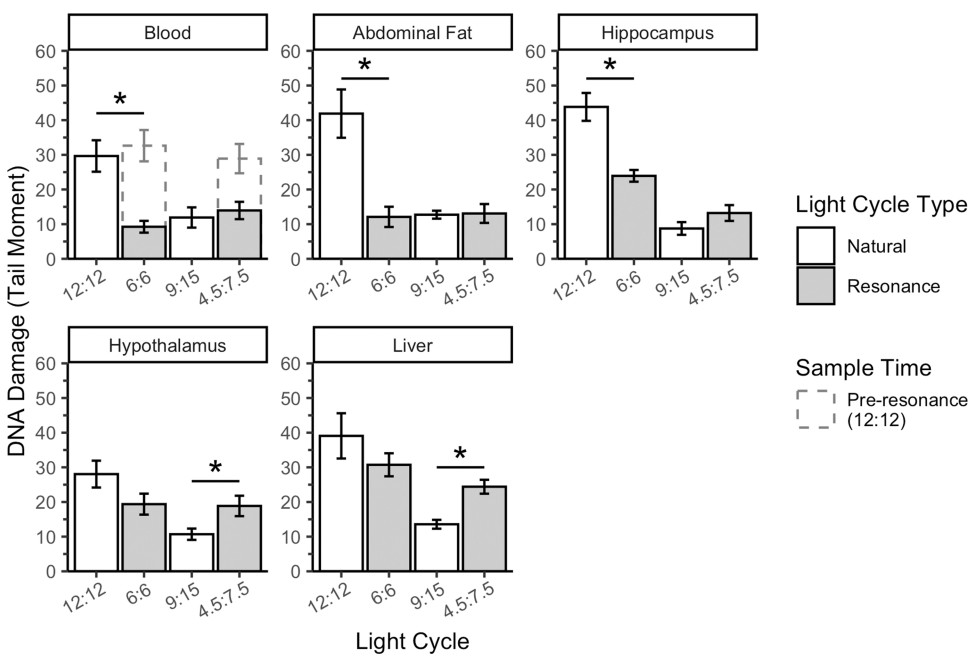

**Figure 3** **Resonance light cycles alter DNA damage in house sparrows in a tissue specific and light cycle specific manner.** A longer tail (and larger tail moment) means more DNA damage. Key comparisons are between resonance and natural light cycles within the same amount of total light in 24 h (12:12 vs. 6:6 and 9:15 vs. 4.7:7.5). Dotted bars in the blood panel represent pre-resonance DNA damage in the same birds, which was a light cycle of 12:12. "Natural" DNA damage data taken from *Beattie et al. (2022a)*. * indicates $p < 0.05$ between key comparisons. N = 9M per resonance group; $n = 5$ (3F:2M) and $n = 8$ (3F:5M) for the 12:12 and 9:15 natural groups, respectively. Plotted values are means ± standard error.

Resonance birds' body weight remained constant through the treatment (Fig. 4A; $F_1 = 16.00$, $p = 0.88$) and did not differ with treatment group (main effect of light cycle: $F_1 = 31.11$, $p = 0.27$; interaction: $F_1 = 16.00$, $p = 0.79$). Baseline corticosterone significantly increased upon treatment with resonance light cycles (Fig. 4B; $F_1 = 13.88$, $p = 0.01$) and this effect was consistent across treatment groups (main effect of light cycle: $F_1 = 26.21$, $p = 0.75$; interaction: $F_1 = 14.34$, $p = 0.94$). Conversely, blood DNA damage significantly decreased upon treatment with resonance light cycles (Fig. 4C; $F_1 = 15.29$, $p = 0.0001$) and was again consistent across treatment groups (main effect of light cycle: $F_1 = 29.88$, $p = 0.46$; interaction: $F_1 = 15.29$, $p = 0.21$).

Resonance birds had enlarged testes (Fig. 5)—the 4.5L:7.5D birds testes weighed 117.9 mg ± 32.00 mg and the 6L:6D birds testes weighed 344.19 mg ± 65.96 mg (mean ± standard error). Testes mass can reach 500 mg during the breeding season or can be as low as 5 mg during the nonbreeding season (*Hegner & Wingfield, 1990*). The resonance birds in this study were caught in August (6L:6D group) and October (4.5L:7.5D group), when breeding anatomy would not be expected. The 6L:6D birds had significantly larger testes than the 4.5L:7.5D birds ($t = 3.09$, $p = 0.01$).

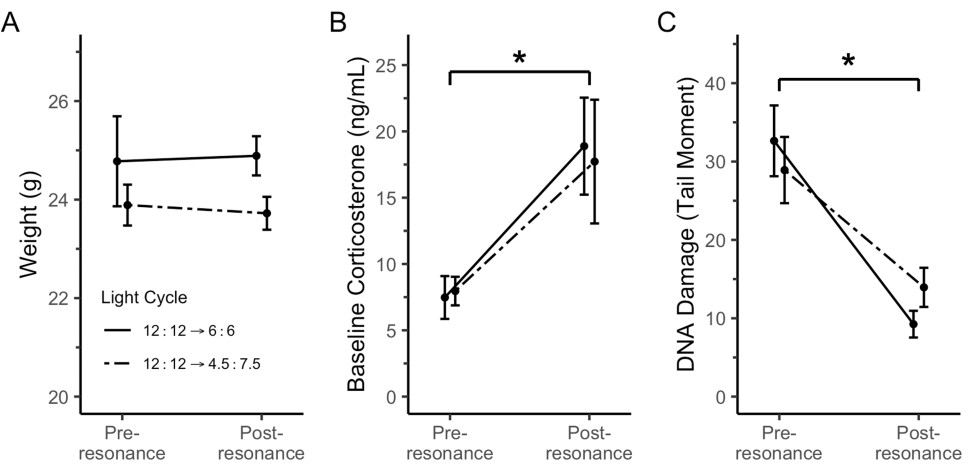

**Figure 4** **Both resonance light cycles did not change body weight, but did cause an increase in baseline corticosterone and a decrease in erythrocyte DNA damage.** All house sparrows were on a 12:12 light cycle at the pre-resonance timepoint. Post-resonance samples were taken four weeks after switching to a resonance light cycle. * indicates $p < 0.05$ over time. N = 9M per group. Plotted values are means ± standard error.

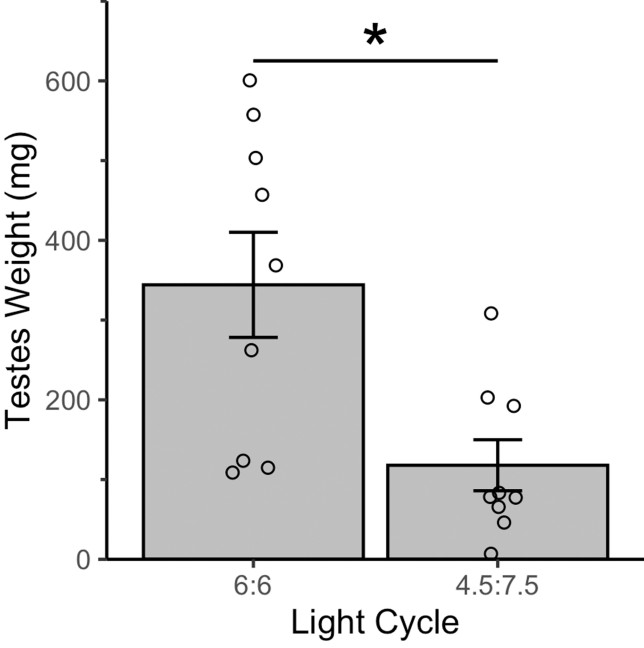

**Figure 5** **All house sparrows had enlarged testes after four weeks of a resonance light cycle.** The 6:6 birds had significantly larger testes than the 4.5:7.5 birds. * indicates $p < 0.05$. N = 9M per group. Plotted values are means ± standard error.

## DISCUSSION

The first part of the present study was to determine if there is a daily rhythm of DNA damage in house sparrows. We found that while uric acid, a potent antioxidant, does show a daily rhythm, DNA damage in blood does not (Fig. 2). An argument could be made for expecting DNA damage to be higher at night, when glucocorticoids peak (*Rich & Romero, 2001*), which is the case for oxidative damage to DNA in great tits *(Parus major)* (*Coughlan, Sadowska & Bauchinger, 2023*). However, in addition to causing DNA damage, glucocorticoids also upregulate genes involved in damage signaling and repair (*Flint et al., 2007*). Conversely, melatonin, another endogenous antioxidant (*Tan et al., 2010*), peaks at night (*Hau & Gwinner, 1994*), so perhaps we should expect DNA damage levels should be highest during the day. However, data from the present study (Fig. 2B), show that a different antioxidant, uric acid, peaks during the day, likely due to exogenous uric acid from food (*Hafez, Abdel-Rahman & Naguib, 2017*). In fact, studies in humans show that different components related to oxidative stress peak during the day just as often as they peak at night (reviewed in *Wilking et al., 2013*). The different daily peaks of variables related to DNA damage may even out, so that damage is held approximately constant under baseline conditions.

In the second experiment, we had intended to use the resonance photoperiods to separate the number of light hours in a 24-hour period from one light-dark cycle (Fig. 1). Within the tissue DNA damage data (Fig. 3), there is conflicting evidence that the seasonal patterns are due to the total number of light hours. Only the treatment groups with 9 total hours of light (4.5:7.5 and 9:15) support that hypothesis because there was no difference in DNA damage between the natural and resonance birds. If the total amount of light in a 24-hour period is what determines the level of DNA damage, then it should not matter how those hours are distributed, such as with 9 h straight of light or broken into two 4.5-hour periods (Fig. 1). On the contrary, the treatment groups with 12 total hours of light (6:6 and 12:12) do not support this hypothesis, as the 6:6 and 12:12 birds had different levels of DNA damage, despite the total hours of light in a 24-hour period being the same. However, our resonance treatment may have had unintended confounding effects on the birds' physiology. We were surprised to find that after four weeks of resonance photoperiods, the 6:6 and 4.5:7.5 birds had enlarged testes (Fig. 5), suggesting they had been photostimulated and were preparing to breed (*Hegner & Wingfield, 1990*). This idea is supported by an increase in baseline corticosterone after four weeks of resonance photoperiod (Fig. 4B), which is another hallmark of the breeding season (reviewed in *Romero, 2002*). Breeding physiology is induced by the day length becoming longer (*Sharp, 2005*), which did not occur with the resonance photoperiods. Within one light/dark cycle, all of the resonance birds had a decrease in day length (12 h of light to either 6 or 4.5 h) and within a 24-hour period, the birds' day length didn't change (12 h of light to 12 h of light, split into two 6-hour periods) or decreased (12 h of light to 9 h of light, split into two 4.5-hour periods). However, even though photoperiod did not increase, it appears as though the birds interpreted the light cycles as if it had. Because of the evidence of photostimulation and breeding physiology, we

believe we inadvertently subjected the resonance birds to a skeleton photoperiod (*Anushi & Bhardwaj, 2006*).

Where a typical 12-hour photoperiod would involve 12 h of continuous light, a skeleton photoperiod might involve 1 h of light, followed by 10 h of darkness, and then 1 h of light again. If the 1 h periods of light occurred at critical times, the animal would interpret the 1L:10D:1L as 12 h of continuous light. This phenomenon has previously been documented in house sparrows (*Binkley, Mosher & Reilly, 1983*). Observationally, the birds in the present study were not active during the mid-day scotophase in the resonance light cycle (*Binkley & Mosher, 1985*), however if the pineal gland and hypothalamic pacemaker were stimulated at critical timepoints, the 6:6 and 4.5:7.5 resonance light cycles could have been interpreted as 18L:6D and 16.5L:7.5D, respectively. In this case, the day lengths would have been interpreted as longer than at the start of the resonance experiment, thus explaining the induction of breeding physiology. As noted above, house sparrows rarely breed in captivity, so perhaps a regimen as extreme as a resonance/skeleton photoperiod was strong enough to induce breeding physiology. The length of these potential skeleton photoperiods matches the testes size with the 6:6 birds having larger testes than the 4.5:7.5 birds (Fig. 5). Similar results have been found in Japanese quail (*Coturnix japonica*), with testes weight, LH, and FSH being positively associated with skeleton photoperiod length (*Follett & Milette, 1982*). Data from the present study, along with other avian skeleton photoperiod research (for example, *Binkley, Mosher & Reilly, 1983*; *Binkley & Mosher, 1985*; *Follett & Milette, 1982*) provides further support in birds for the External Coincidence Model of how animals measure day length (*Bentley, 2009*; *Bünning, 1936*). This model posits that animals are insensitive to light during the photoperiod, but if light occurs during what should be the scotophase, the photoperiodic response will be initiated.

Given the onset of the breeding physiology, based on testes growth (Fig. 5; *Hegner & Wingfield, 1990*) and baseline corticosterone increase (Fig. 4B; reviewed in *Romero, 2002*), it is surprising that blood DNA damage decreased (Fig. 4C). Other systems, such as the immune system (*Folstad & Karter, 1992*), are downregulated during breeding, likely because of tradeoffs related to the increased energy expenditure of preparing to breed. Though the data are mixed, oxidative stress may increase with reproduction (*Speakman & Garratt, 2014*; *Wiersma et al., 2004*), which could lead to an increase in DNA damage. Furthermore, free-living house sparrows have higher DNA damage during the breeding season, which is mimicked in captivity with an equivalent photoperiod (*Beattie et al., 2022a*). Although we did not measure energy expenditure while in captivity, the resonance birds in the present study appear to be preparing to breed without incurring the cost of DNA damage typically associated with longer day lengths and/or reproduction.

An alternative explanation for the increase in baseline corticosterone (Fig. 4B) throughout this experiment is that the resonance light cycles were a chronic stressor for the birds. While increased corticosterone does not necessarily mean an animal is chronically stressed (*Romero & Beattie, 2021*), the fact that body weight does not change (Fig. 4A; *Dickens & Romero, 2013*) and that DNA damage decreases (Fig. 4C; *Gormally et al., 2019b*) suggests that these birds were likely not experiencing chronic stress.

We were surprised to find tissue-specific differences when comparing the resonance treatment groups to the natural groups, because seasonal patterns in DNA damage were uniform across tissues (*Beattie et al., 2022a*). In the 6:6 resonance group, blood, abdominal fat, and hippocampus DNA damage was significantly lower than in the 12:12 group, but damage in the hypothalamus and liver was not. Specifically, we would have expected the hypothalamus and hippocampus to respond in a similar manner, as both tissues are involved in circadian processes (*Abraham, Albrecht & Brandstätter, 2003*; *Smulders, 2017*), however as discussed above, DNA damage may not be related to the circadian or daily rhythm. Contrary to the 6:6 treatment group, the 4.5:7.5 treatment group had lower DNA damage in the hypothalamus and liver than the 9:15 group. It is unclear why resonance light cycles would have the opposite effect on DNA damage depending on the tissue analyzed, but, with the reproduction data, is further evidence that different systems of the house sparrow can respond differently to the same stimulus.

## CONCLUSIONS

In conclusion, although our experiments were not able to fully elucidate the mechanism of seasonal DNA damage in the present study, we were able to show that it is not a direct reflection in the amount of light hours. If this had been the case, we would have expected the corresponding natural and resonance light cycles to have the same levels of DNA damage. We also showed that there is no daily rhythm of DNA damage. Baseline corticosterone increased while DNA damage decreased throughout the duration of the resonance experiment, indicating that corticosterone is not a driver of seasonal patterns in DNA damage, as it is during acute stress (*Beattie et al., 2024*; *Flint et al., 2007*), potentially due to receptor dynamics. Finally, we demonstrated a potential uncoupling of the reproductive system and seasonal DNA damage by treating birds with resonance photoperiods that may have been interpreted as skeleton photoperiods. Some fruitful avenues for future research could include directly manipulating activity levels, altering photoperiods outside of the critical timepoints that trigger a response to a skeleton photoperiod, and further untangling the link between glucocorticoid exposure and DNA damage.

## ACKNOWLEDGEMENTS

We thank Emma Kudej, Lauren Kole, Bradley Pedro, Rachel Ricco, Paul Jerem, and Denyelle Kilgour for their help taking blood samples as well as Simone Meddle for advice on the resonance photoperiods. We also thank the Tufts University animal care staff.

### Funding

This work was supported by the National Science Foundation (IOS-1655269). The funders had no role in study design, data collection and analysis, decision to publish, or preparation of the manuscript.

## Grant Disclosures

The following grant information was disclosed by the authors:
National Science Foundation: IOS-1655269.

## Competing Interests

The authors declare there are no competing interests.

## Author Contributions

- Emma Rosen conceived and designed the experiments, performed the experiments, analyzed the data, prepared figures and/or tables, authored or reviewed drafts of the article, and approved the final draft.
- Lily Mikolajczak conceived and designed the experiments, performed the experiments, analyzed the data, prepared figures and/or tables, authored or reviewed drafts of the article, and approved the final draft.
- Ursula K. Beattie conceived and designed the experiments, performed the experiments, analyzed the data, prepared figures and/or tables, authored or reviewed drafts of the article, and approved the final draft.
- L. Michael Romero conceived and designed the experiments, performed the experiments, authored or reviewed drafts of the article, and approved the final draft.

## Animal Ethics

The following information was supplied relating to ethical approvals (i.e., approving body and any reference numbers):

All experiments were approved by the Tufts University Institutional Animal Care and Use Committee.

## Data Availability

The raw data and the R code are available in the Supplemental Files.

## Supplemental Information

Supplemental information for this article can be found online at http://dx.doi.org/10.7717/peerj.18375#supplemental-information.

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
