# Peer review of "House sparrows do not show a diel rhythm in double-strand DNA damage in erythrocytes"

_PeerJ, doi:10.7717/peerj.18375_

## Round 0.1 · original submission · Major Revisions

Dear Dr. Beattie,

This paper was evaluated by three experts in the field who appreciated the study and suggested it should be published. At the same time, they raised several well-informed comments that must be taken into consideration in a revision. These comments are either requests for more clarity on methods and hypotheses or more clarity on the introduction. All these comments are very clearly presented and require no additional explanation. Please revise the manuscript considering all the suggestions given by all three reviewers.

Reviewer 1 ·

Basic reporting

Figure captions- Several important details are missing from the figure captions. In each caption, please include sample size, species name, and a statement of what the markers represent (e.g. means+/- SEM).

For Figure 1, the x-axis would be improved if a HH:MM format was used- e.g. 05:00 vs. 5. It would make more sense to include major units every 4 rather than 5 hours, so this would line up with plotted points. Finally, please include minor tick marks on both x- and y-axes so readers can better interpolate plotted data points (for all figures). Finally, in the Figure 1 caption, it should say “Different letters in panel B indicate…”

For Figure 2, it would be helpful to remind the reader in the caption that a longer tail means more DNA damage.

Experimental design

My biggest concern is that there is a lack of information about what time of year sparrows were captured for both the diel rhythm and resonance light cycle experiments. Additionally, it would be nice to know if the month of capture in this study was similar to the former study cited in this paper (Beattie et al., 2022a). I’m guessing birds were in non-breeding condition when captured for the resonance photoperiod experiment, as some of them then developed into breeding condition during the experiment, and that birds were caught during the late spring or early fall for the diel rhythm experiment, as the natural light cycle was close to 12L:12D. If this was the case, then it is likely not appropriate to compare them, as time of year at capture may have affected the birds transition into captivity and ability to adjust to a new light cycle regimen. It’s also difficult to say whether it’s appropriate to compare these findings with Beattie et al. (2022a), as that manuscript also lacked information in regards to when birds were captured from the wild before being placed on different photoperiods. There may be significant effects caused by transitioning from a wild long-day cycle to a captive short-day cycle vs. transitioning from a wild short-day cycle to a captive long-day cycle. In summary, I would appreciate some more details concerning the time of year sparrows were caught in this study, and if time of year was different between the two experiments in this study or from Beattie et al. (2022a)- I’d appreciate justification for why this would be biologically appropriate for testing the proposed hypotheses, as well as text in the discussion addressing how findings could be influenced by time of capture.

The authors do a nice job of discussing how the resonance photoperiods did not work as expected. Therefore, this manipulation caused birds to make different interpretations of their photoperiodic schedule, and therefore does not adequately test the hypotheses in this study, as it did not provide a suitable way to change the total number of active hours within a 24-h day. I think the framework the authors use in their introduction is still fine, and the discussion is also fine. But I’d appreciate more qualification in both the abstract and the summary. Instead of saying “neither hypothesis was supported,” it would be more appropriate to state “our experimental manipulation was inadequate to effectively test our hypotheses, as…”. Given this, the current title of the manuscript is misleading (also because the amount of time spent active was not actually quantified). I’d suggest revising the title to reflect the most solid finding of this study, where sparrows showed no diel rhythms in erythrocyte DNA damage.

Another concern I have regards the amount of time sparrows were given to acclimate to captivity- in Beattie et al. (2022a), sparrows were given 4 weeks, but in the current study birds were given a minimum of 2 weeks to acclimate. Could differences between this study and the prior study be due to this? Is there prior knowledge of how DNA damage changes over the period of acclimation to captivity? Is the stress of transitioning to captivity more intense during summer vs. winter, and if so, does it take longer for birds to acclimate in one season vs. another? I’d appreciate more justification regarding this in both the methods and possibly the discussion section. I wondered if the drop in DNA damage from pre-resonance to 4 weeks later could just be due to the fact that birds are habituating to captivity, and that 2 weeks are not sufficient acclimation time?

It also appears that Beattie et al. (2022a) had a more-or-less even mix of male and female sparrows in their study, while the current study predominantly uses males. Could this be another reason for different findings between the studies? I’d appreciate justification for why few females were used in the current study.

It seems that transition into reproductive condition via the resonance photoperiod supports the “external coincidence model,” as detailed by Bentley, G.E. 2009. “Photoperiodism and Reproduction in Birds”, in Photoperiodism: The Biological Calendar (ed. R.J. Nelson). Please include more discussion relating findings from this study with those of other birds.

While high corticosterone levels during acute stress seem to be related with increased DNA damage (Beattie et al., 2024), the present study found, via the resonance experiment, that higher corticosterone levels coincided with lower DNA damage levels. Could this perhaps suggest that cort-induced DNA damage only occurs via GR activation (which happens at stress-induced cort levels, like in Beattie et al., 2024) but not via MR activation (which is more typical at baseline levels, like in the current study)?

Validity of the findings

See first comment, above, in Experimental Design section.

Additional comments

L47: Perhaps temperature could be included in this list of environmental variables?

L81: Insert “Animal” in between “Institutional” and “Care”

L90: Please include the location where birds were housed. Also, please state the average temperature birds experienced in captivity, and the light intensity they experienced (past studies have found that light intensity affects photostimulation in birds).

L96: Please state the average or maximum amount of blood collected.

L98: Please state at what time lights turned on and off for this experiment- it would be helpful to know if, for example, 8am was when lights came on, or 1 hr after lights came on, etc. I’m guessing based on Fig 1 that lights came on at 8am and turned off at 8pm.

L103: How much blood is needed for this measurement?

L158: I’d appreciate more details, here. How much plasma was typically used per sample and how much was it diluted before assaying? Was plasma extracted before running the EIAs, and if so, please briefly describe these methods. Finally, please state the intra- and inter-assay CVs, and state whether samples from different treatments were evenly distributed between plates.

L218: Is there a comparison group for testes? Meaning, do birds captured at the same time of year from the wild have lower testes mass than 4.5L:7.5D birds?

L258: It might be nice to further explain how skeleton photoperiods work, as some readers may not have a strong background in chronobiology.

L269: “…the data are mixed” not the “the data is mixed”

L272: Energy budgets likely change a lot over the breeding season, and likely differ between the sexes, as there are large differences in the types of activities required for different reproductive stages including mating, nest building, egg formation, incubation, and offspring rearing. For captive birds exposed to long daylengths, the only major change is recrudescence of the gonads… could this really result in a substantial increase in energy use, and thus increase oxidative stress and DNA damage?

Reviewer 2 ·

Basic reporting

Here, the authors report on two studies that ask why DNA damage appears to have a seasonal pattern in the house sparrow. The authors use clear, unambiguous language throughout and supply readers with clear figures and raw data. In general, the manuscript is in good shape; however, I would encourage the authors to adjust their introduction and discussion to include a broader context for their study – e.g., overall importance of DNA damage, expectations of tissue specificity, caveats and other confounding factors, etc. Currently, the lack of context in some regard waters down the conclusions of the paper, which currently, emphasize more the potential skeleton photoperiod rather than the main question they are asking. With a few adjustments, the authors’ manuscript should be ready for publication in this journal.

Experimental design

This manuscript fits the aims and scope of PeerJ and the experiment appears to be performed as a high technical and ethical standard that can be replicated according to the results. My main critique is that the manuscript may benefit from a broader context in the introduction and discussion (see Basic Reporting & specific comments sections), which would better cement the relevance of the research topic and how it fills a knowledge gap. For example, what is the importance of DNA damage to the individual, evolution, etc.? What are the consequences of tissue-specific DNA damage? What are different natural photoperiods that animals live under, and could this influence their fitness, etc.? What are the consequences of a skeleton photoperiod, and would this be something you could see in wild populations? By the end of the manuscript, it is a little unclear what the punchline and importance are.

Validity of the findings

See above for comments about the conclusions and punchline of the manuscript.

Additional comments

General
1. Throughout the manuscript, the authors rely quite heavily on several jargon terms, which could be more clearly defined early in the manuscript for a more general audience. E.g., “resonance,” “scotophase,” “photophase,” and “skeleton photoperiod.”

Title
1. The title does not fully encompass the authors’ findings nor the general question being asked. In addition, “time spent active” is not the only thing manipulated and at first glance suggests that the authors are measuring time spent active, which I believe they did not.

Abstract
1. L22-23: I encourage the authors to more generally explain these different light cycles, especially given that the reader might not know what a resonance experiment is.
2. L30-32: Can the authors end the abstract with a conclusion more based in their main research question? That this resonance study might have unintended consequences is certainly interesting, but tangential to the main research question. How do these findings apply to broader topics or importance?

Introduction
1. General: The authors do a good job of explaining that DNA damage can be influenced by a variety of factors, including GCs, for example, but something I think could anchor this study is by briefly stating why DNA damage matters? What are the consequences more broadly?
2. L51-56: I understand the two potential hypotheses the authors suggest for seasonal patterns of DNA damage. However, the introduction seemed to lack any caveats about other potential drivers of DNA damage seasonally beyond circadian regulation and activity level. The authors do mention solar radiation, but there seem to be other more relevant factors like reproduction that could be mentioned as well.
3. L57-66: The authors talk about DNA damage regulators that also vary rhythmically, including GCs and antioxidants. I’d love to know more about why the authors chose uric acid, which shows an inconsistent diel rhythm compared to other antioxidants. In addition, it might benefit readers to plant the seeds for why the authors are measuring both CORT and uric acid in these birds. What can that data tell us beyond what the DNA damage assays can?
4. L71-76: I appreciate the authors’ predictions here – they help immensely.

Materials and Methods
1. L91-93: The authors write that six males were housed singly, and then 2 males and 2 females were housed in pairs. I’m a little confused about why this happened, and what the potential consequences are for this protocol, as some (if not all) of the males were in the presence of female conspecifics. Could this have influenced the testes results? Or been more stressful for some birds than others?
2. General: Could a figure or schematic help readers understand the timing of light cycles during the resonance experiment? Not absolutely necessary – just a thought. I think it immediately needs to be made obvious that the total number of hours and chunks of daylight differ between these treatments.
3. General: There is a large assumption to this study, in which birds are actually active during daylight conditions. Yet, the authors do not report any evidence that birds were more active during daylight conditions or whether activity differed between the 4 treatment groups. This has consequences for the interpretation of the results that I think is important, if the authors can give some insight. Obviously, if the data does not exist, it does not need to be redone, but a caveat might be appropriate.
4. L159-162: There is not very much information about the CORT assays in the main text. Is there any additional information the authors should consider including, e.g., citation to a previous publication, inter-plate variation, etc.?

Results
1. L195-199: I found the results section a bit difficult to keep straight, especially for the main model. I might suggest that the authors consider a table instead of reporting test statistics in the main text. They might also include effect sizes or beta values as well.
2. Figure legends: Some appeared a bit sparse in their information.

Discussion
1. In general, I feel like the discussion could more effectively bring in other literature to interpret the authors’ results. For example, again, what other factors could be impacted by resonance light cycles, like sleep cycles and depth, eating patterns, reproduction, etc.? Here or in the introduction, can the authors posit why we might see or expect tissue-specific patterns and/or what the consequences might be? Is there any additional interpretation to be made from the CORT and uric acid data? Are there other examples of skeleton photoperiods happening and/or is there any other relevant information about this? Could there be any caveats about the timing of sample collection that could influence things? Etc. The discussion as is is a little sparse and I’m not super sure what the take-aways are about the main research question of the paper.

Reviewer 3 ·

Basic reporting

All the figures are relevant, but they need more description. It is not clear what the dots/bars represent - averages from raw data or from analysis? It is not written what the error bars represent.
Raw data are provided, but it would be useful to have metadata explaining what is in them.

It is unclear why DNA damage was measured in five tissues and why these tissues were chosen.

Experimental design

The diurnal rhythm experiment was performed on only 2 females and 8 males, and in addition, 6 males were housed separately and two other males were housed with females. This potentially caused a huge variance in the data, and because of the small sample size, it is not possible to control for it statistically. I would like to see the same results analyzed for just the 6 males that were housed separately.

It is not clear why the statistical analyses for the resonance experiment were split into two instead of using the slice option. This helps with statistical power and may show differences in DNA damage, for example for the liver.

Validity of the findings

As the baseline corticosterone increases, it would also be interesting to discuss it in the context of chronic stress caused by resonant light cycles.

---

## Round 0.2 · Minor Revisions

Dear Dr. Beattie,
Thank you for submitting the revision. I am happy to inform you that your paper has been approved by two reviewers for publication after addressing a few minor suggestions. Please address these comments in the final revision.

L121: Please report temperature in C, not F.
L122: Please change the second "photophase" to "scotophase"
L139-142: Please indicate whether all August birds were put on one photoperiod in captivity while all October birds were put on another, or whether you equally distributed Aug/Oct birds between the two experimental photoperiods.
L201-204: Were samples plated in duplicate or triplicate?

Reviewer 1 ·

Basic reporting

No comment.

Experimental design

No comment.

Validity of the findings

No comment.

Additional comments

L121: Please report temperature in C, not F.
L122: Please change the second "photophase" to "scotophase"
L139-142: Please indicate whether all August birds were put on one photoperiod in captivity while all October birds were put on another, or whether you equally distributed Aug/Oct birds between the two experimental photoperiods.
L201-204: Were samples plated in duplicate or triplicate?

Reviewer 2 ·

Basic reporting

I thank the authors for their edits following a previous review. I am happy with the edits and have no further comments.

Experimental design

No comments.

Validity of the findings

No comments.

Additional comments

Great work on the edits! Thanks.

---

## Round 0.3 · accepted · Accept

Dear Dr. Beattie,

Thank you for submitting the revised manuscript to PeerJ.

I am writing to inform you that your manuscript - House sparrows do not show a diel rhythm in double-strand DNA damage in erythrocytes - has been Accepted for publication. Congratulations!